# Bitblasting for Constrained Decorrelation in Tractable Image Modeling

Poorva Garg[1]    Benjie Wang[1]    Oliver Broadrick[1]    Guy Van den Broeck[1]    Todd Millstein[1]

[1]Department of Computer Science, University of California, Los Angeles, California, USA

## Abstract

Probabilistic circuits (PCs) are tractable probabilistic models, enabling exact and efficient computation of many queries. When modeling images with PCs, one key step in the learning pipeline is *decorrelation*. Since RGB channels in pixels are highly correlated in natural images, learning is instead performed on a transformed pixel space with much lower channel-wise correlation, making the learning task easier. However, the transformations are not bijective; there are values in the modeled space not realizable as images in the original space. In particular, probability mass is 'leaked' to such invalid values during learning on the transformed space. Moreover, the resulting model does not enable tractable inference on the original space. We propose to use bitblasting – representing a distribution over complex objects as a distribution over bits – to address these problems. We show that the relationship between the original and transformed spaces can be encoded exactly and succinctly in the structure of the PC, removing the leakage problem, improving modeling performance, and providing a tractable model over the original space. Preliminary empirical results support our approach.

## 1 INTRODUCTION

Probabilistic circuits (PCs) are deep generative models whose structure enables efficient, exact probabilistic inference [Choi et al., 2020, Darwiche, 2021], enabling applications like controllable generation and model alignment [Liu et al., 2024b, Zhang et al., 2024, Yidou-Weng et al., 2025]. Much research on the architecture and implementation of PCs has scaled them to increasingly complex distributions like natural language and images [Liu et al., 2024a, Zhang et al., 2025b, Wang and Van den Broeck, 2025].

When PCs are used to model images, one important technique is *decorrelation*. The RGB channels in natural images are highly correlated, but there exist simple hand-crafted linear transforms that can almost entirely remove the correlation between channels. It has been observed empirically that it is easier to learn PCs on the decorrelated space, and this technique is standard practice in the literature [Liu et al., 2023a,b, Gala et al., 2024, Liu et al., 2024a]. However, this approach has two major drawbacks. First, the pixel channels in images only take values up to some bitdepth (i.e. 8 bits for each channel), but more bits are needed in each channel to capture the full range of the transformation. Consequently, some values in the transformed space are not realized by actual RGB images, and probability mass is inevitably 'leaked' to these values. Second, the resulting models from learning on the decorrelated color space only model that space tractably and not the original RGB space.

We propose to address these concerns by representing a distribution over images as a distribution over the *bits* in the binary expansion of each channel value. Reducing probabilistic models of complex objects to distributions over bits has proven useful in other probabilistic inference contexts and is known as *bitblasting* [Garg et al., 2024, Cao et al., 2023, Sladek et al., 2025]. We show that in such a representation, a linear transformation and the constraint introduced by it can be efficiently encoded *exactly* in the PC, ensuring that no mass is leaked. Moreover, it enables us to obtain a tractable model in the original RGB space without sacrificing accuracy. Empirically, we observe that constraining the decorrelated space leads to a significant improvement in likelihood for PCs modeling ImageNet patches. Additionally, by incorporating the constraint directly into the PC structure, we are able to optimize the PC parameters further in the constrained function space.

We provide background on PCs and image data decorrelation in section 2 and describe the problem of leakage in section 3. We present our approach based on bitblasting in section 4, describe our experiments and results in section 5, and conclude in section 6.

*Accepted for the 8th Workshop on Tractable Probabilistic Modeling at UAI  (TPM 2025).*

| | Bits | BPD | Pairwise correlations |
|---|---|---|---|
| **RGB** | (8, 8, 8) | 7.01 | 0.92, 0.81, 0.92 |
| **YCoCg-L** | (10, 9, 10) | 6.86 | 0.00027, 0.19, −0.00049 |
| **YCoCg-R** | (8, 9, 9) | 6.80 | −0.00027, 0.019, −0.00051 |
| **YCoCg-mod** | (8, 8, 8) | 6.83 | 0.15, 0.11, 0.068 |

Table 1: Common color transforms with (i) the number of bits needed to be lossless; (ii) test bits-per-dimension for a PC trained on single pixels from CIFAR-10 (lower is better) (iii) pairwise correlations among channels (lower is better).

## 2 BACKGROUND

**Probabilistic circuits.** Probabilistic circuits [Choi et al., 2020] are a general class of tractable probabilistic models that express high-dimensional distributions as computation graphs.

**Definition 1** (Probabilistic Circuit). *A probabilistic circuit (PC) $\mathscr{A} = (\mathcal{G}, \boldsymbol{\theta})$ represents a joint probability distribution over random variables $\boldsymbol{X}$ through a rooted directed acyclic (computation) graph (DAG), consisting of sum ($\oplus$), product ($\otimes$), and leaf nodes (L), parameterized by $\boldsymbol{\theta}$. Each node $t$ represents a probability distribution $p_t(\boldsymbol{X})$, defined recursively by:*

$$p_t(\boldsymbol{x}) = \begin{cases} f_t(\boldsymbol{x}) & \text{if } t \text{ is a leaf node} \\ \prod_{c \in ch(t)} p_c(\boldsymbol{x}) & \text{if } t \text{ is a product node} \\ \sum_{c \in ch(t)} \theta_{t,c} p_c(\boldsymbol{x}) & \text{if } t \text{ is a sum node} \end{cases}$$

*where $f_t(\boldsymbol{x})$ is a univariate input distribution function (e.g. Gaussian, Categorical), we use $ch(t)$ to denote the set of children of a node $t$, and $\theta_{t,c}$ is the non-negative weight associated with the edge $(t,c)$ in the DAG. We define the scope of a node $t$ to be the variables it depends on. The function represented by a PC, denoted $p_{\mathscr{A}}(\boldsymbol{x})$, is the function represented by its root node; and the size of a PC, denoted $|\mathscr{A}|$, is the number of edges in its graph.*

The key feature of PCs is their *tractability*, i.e., the ability to answer queries about the distributions they represent exactly and in polynomial time. For example, the commonly assumed properties of smoothness and decomposability ensure that a PC computes a multilinear polynomial in the input distributions, a sufficient condition for efficient answering of marginal and conditional inference queries [Broadrick et al., 2024, 2025].

**Color Transforms.** Pixels in digital images are typically represented using three color channels – red, green, and blue – for reasons ultimately relating to human physiology. However, in a number of digital settings, it is useful to represent pixels in other, transformed color spaces. For example, the YCoCg family of color spaces separate a luminance (Y) from two chroma channels. In natural images, RGB color channels are highly correlated; whereas YCoCg color spaces statistically decorrelate the three channels. Moreover, the transformation from RGB space to a YCoCg space is a simple, linear map. These properties, in particular decorrelation, are exploited in various settings, for example in data compression [Goyal, 2002, Agawane, 2008, Prativadibhayankaram et al., 2024]. In the literature on PCs, it has been observed that application of such decorrelating color transforms significantly boosts image modeling performance (as measured by log-likelihood) compared to directly modeling on RGB datasets [Liu et al., 2023a,b, Gala et al., 2024].

## 3 PROBLEMS WITH DECORRELATING COLOR CHANNELS IN IMAGES

For RGB pixels with bitdepth 8 for each channel, Table 1 lists several commonly used transforms. However, when applying a linear transform with inputs of fixed bitdepth, extra bits are required for these transforms to be injective (and thus lossless). For example, the well known YCoCg (YCoCg-L in Table 1) transform is simple and exhibits excellent channel decorrelation [Malvar and Sullivan, 2003b]. Indeed, it was found by taking the linear transformation with maximum decorrelation on a dataset of natural images and rounding its entries [Malvar et al., 2008]. The YCoCg transform is given as follows:

$$\begin{pmatrix} Y \\ Co \\ Cg \end{pmatrix} = \begin{pmatrix} 1/4 & 1/2 & 1/4 \\ 1/2 & 0 & -1/2 \\ -1/4 & 1/2 & -1/4 \end{pmatrix} \begin{pmatrix} R \\ G \\ B \end{pmatrix} \qquad (1)$$

Given integers $R$, $G$, $B$ in the range $[0, 255]$, $Y$ takes values in the range $[0, 255]$ at an interval of $0.25$, i.e., $Y$ can take $1024$ different values, requiring 10 bits to be represented losslessly. In total, this encoding requires 5 additional bits increasing the size of the modeled space by $32\times$. As a result, not all values in the resulting YCoCg space can be mapped back to RGB values. For example, mapping back $(Y, Co, Cg) = (0.25, -0.5, 0.25)$ results in $(R, G, B) = (-0.5, 0.5, 0.5)$ which are not integers and hence, not valid values for RGB channels.

YCoCg requires fixed-point arithmetic to be represented losslessly which is not ideal. To avoid this, the YCoCg-R transform is often used that computes a close approximation to the YCoCg transform. It is a reversible transform under integer operations [Malvar and Sullivan, 2003a] and also exhibits excellent decorrelation (see appendix B for details). However, it still requires 2 additional bits to be injective. Finally, the YCoCg-mod transform [Strutz and Leipnitz, 2015] uses modular arithmetic to avoid the need for additional bits, but has worse decorrelation.

In summary, each known transform either requires extra bits for a lossless representation (and so a learned model on the transformed space leaks probability mass), or does not decorrelate the channels as effectively.

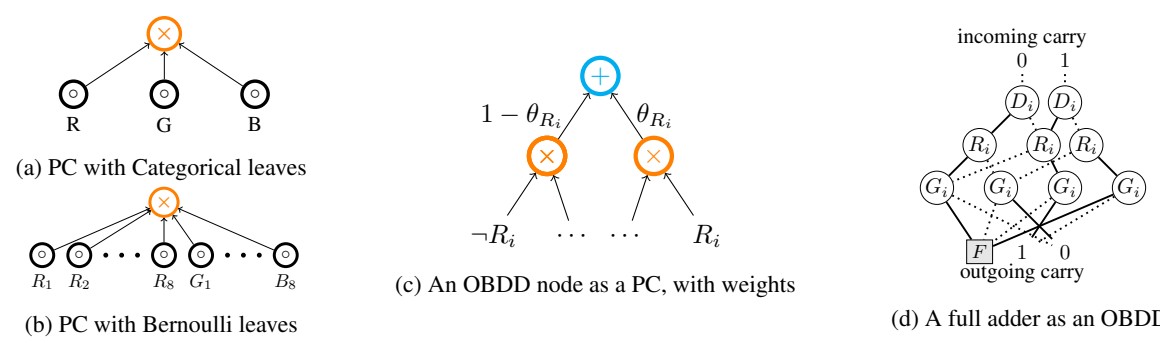

(a) PC with Categorical leaves

(b) PC with Bernoulli leaves

(c) An OBDD node as a PC, with weights

(d) A full adder as an OBDD

Figure 1: Illustration of color transform constraints and PC components.

# 4 BITBLASTING TO LEVERAGE DECORRELATION

As shown in the previous section, modeling images in the decorrelated space leaks probability mass to values that cannot be realized in the original RGB space. Therefore we would like to constrain our model of the transformed space so that such 'invalid' values are always assigned zero mass. Doing so by enumerating all $2^{24}$ possible values is infeasible, and so we seek a compact encoding of the constraint that preserves tractability of the model.

## 4.1 REPRESENTING LINEAR TRANSFORMS USING BINARY REPRESENTATION

In the current state-of-the-art PCs that model images, each channel is represented using leaves with a categorical distribution. Our key insight is that if each channel is instead modeled at the level of its binary representation, imposing the constraints arising from the linear transformation becomes tractable. Specifically, the constraint arising from a linear transformation can be encoded with a small Ordered Binary Decision Diagram (OBDD) [Wegener, 2000] (which can be efficiently transformed to a deterministic, structured decomposable PC [Amarilli et al., 2024]).

To illustrate the approach, consider the case of adding two channels:

$$D = R + G$$

Let R be represented using bits $[R_1, R_2, \ldots, R_8]$ and similarly for G and D. Then, to represent the above linear transform, we need a circuit that encodes the constraint that for any index $i$, the $i$-th bit of the output channel $D$ equals the $i$-th bit of the sum of R and G. Following the grade school algorithm for addition, let $c_i$ be the incoming carry bit for adding the $i$-th bit of R and G. Then, the following holds:

$$D_i \iff c_i \oplus R_i \oplus G_i$$
$$c_{i+1} \iff (R_i \wedge G_i) \vee (c_i \wedge (R_i \oplus G_i))$$

Notice that, given the incoming carry $c_i$, the constraint for the $i$-th bit is independent of all bits with index less than $i$.

| | Include RGB | Exclude RGB |
|---|---|---|
| **YCoCg-L** | 909 | 474 |
| **YCoCg-R** | 1905 | 799 |

Table 2: OBDD sizes for validity constraints YCoCg $\leftrightarrow$ RGB for different color transforms; columns indicate whether the RGB variables are marginalized out or not.

This means that the OBDD for the addition of two channels, with the variable order where bits are interleaved[1], scales linearly in the number of bits. Figure 1d shows the OBDD component for the $i$-th bit of the constraint.

Similar to how bitblasting leads to a compact adder, bitblasting the color channels in the original and transformed space leads to an OBDD that succinctly represents the constraint arising from linear transformation for both YCoCg-L and YCoCg-R. Table 2, shows the resulting OBDD sizes for YCoCg-L and YCoCg-R.

## 4.2 INCORPORATING CONSTRAINTS IN PC TRAINING

Given an OBDD representing the constraint, we construct pixel-level probabilistic circuits $\mathscr{A}_{\text{constraint}}$ satisfying the validity constraint. Specifically, OBDDs can be rewritten as deterministic, decomposable PCs with right-linear structures [Amarilli et al., 2024]. Each node in the OBDD corresponds to the circuit structure shown in Figure 1c. By associating learnable weights to the sum node edges, one can parameterize a probability distribution over only valid color YCoCg channel values. Furthermore, by including the $R, G, B$ variables, we retain a tractable model over those color channels as well. To incorporate this into a tractable image-level PC, nodes modeling a pixel can be replaced by $\mathscr{A}_{\text{constraint}}$.

**Parameter Initialization** In preliminary experiments of learning constrained image-level PCs using the EM algorithm [Dempster et al., 1977, Peharz et al., 2016], we found

---

[1] Following the order $R_1, G_1, R_2, G_2, \ldots, R_n, G_n$.

| | Categorical | | | Binary (no constraint) | | | Binary (constraint) | | |
|---|---|---|---|---|---|---|---|---|---|
| | Train | Test | #Par | Train | Test | #Par | Train | Test | #Par |
| **RGB** | 6.90 | **7.01** | 7690 | 7.15 | 7.24 | 250 | - | - | - |
| **YCoCg-L** | 6.69 | 6.86 | 25610 | 7.08 | 7.20 | 300 | 6.66 | **6.81** | 11050 |
| **YCoCg-R** | 6.64 | **6.80** | 12810 | 6.72 | 6.86 | 270 | 6.67 | 6.83 | 17910 |
| **YCoCg-mod** | 6.67 | **6.83** | 7690 | 6.70 | 6.85 | 250 | - | - | - |

Table 3: Modeling CIFAR pixels (1x1 patch) in different color transforms using different representations of the channels.

| | Categorical | | | Binary (no constraint) | | | Binary (constraint) | | |
|---|---|---|---|---|---|---|---|---|---|
| | Train | Test | #Par | Train | Test | #Par | Train | Test | #Par |
| **RGB** | 5.77 | 5.76 | 46,139,392 | 5.74 | 5.74 | 26,216,448 | - | - | - |
| **YCoCg-L** | 5.92 | 5.89 | 42,993,664 | 5.94 | 5.92 | 27,265,024 | - | - | - |
| **YCoCg-R** | 5.61 | 5.60 | 45,090,816 | 5.59 | 5.58 | 28,313,600 | 5.55 | **5.52** | 58,688,512 |
| **YCoCg-mod** | 5.58 | 5.57 | 46,139,392 | 5.55 | 5.55 | 27,265,024 | - | - | - |

Table 4: Image modeling results on $4 \times 4$ ImageNet32 patches; results given in bpd (bits-per-dimension, lower is better).

that random initialization of weights led to worse performance than the corresponding *unconstrained* PC, indicating a failure of the learning algorithm to find the optimal parameters. Therefore, we instead initialize the constrained PC using the parameters of a trained unconstrained PC. Specifically, we first learn an unconstrained PC where each channel of every pixel is represented in binary and each bit has a Bernoulli distribution. Figure 1b shows the structure for each pixel in the resulting PC. Note that this reduces the number of parameters per channel compared to a Categorical leaf (from $255$ to $8$), though we show in Section 5 that the performance of the resulting PC remains competitive. We then transfer the parameters from the learned unconstrained PC to the constrained PC. For each input Bernoulli distribution, we take its learned parameter and use it to initialize the weights of the corresponding sum node in the constrained PC as shown in Figure 1c. This is equivalent to multiplying the circuit shown in Figure 1b with the constraint, which can be done tractably due to the structure of these circuits [Shen et al., 2016, Vergari et al., 2021, Wang et al., 2024, Zhang et al., 2025a]. This zeros out the invalid values while maintaining the likelihood of the valid ones. Finally, we normalize the constrained PC to complete the process of initializing its parameters. This immediately provides an increase in likelihood compared to $p_{\mathscr{A}}$ simply by redistributing the leaked probability mass. We can further improve the performance by continuing the learning process for the constrained PC from this initialization.

## 5 EXPERIMENTS

We implemented our approach and performed preliminary experiments on the CIFAR-10 [Krizhevsky et al., 2009] and ImageNet32 [Deng et al., 2009] datasets. Our primary research question is to examine the extent to which constraining the input decorrelated space to realizable values

helps the performance of PCs in modeling images.

Table 3 reports results of our experiments of modeling the distribution of single pixels in the CIFAR dataset. The pixel distributions were modeled using 10-component mixture of distributions on the input channels. In these experiments, the constrained circuit was learned using random initialization. As expected, the test bits-per-dimension (BPD) deteriorates when we replace categorical distributions with binary representation since there are fewer parameters, but it improves significantly once we apply the constraint. In the case of YCoCg-L transform, the constrained circuit even outperformed its counterpart with categorical leaf distributions as there is no leakage to invalid values.

Table 4 reports results of our experiments of modeling the distribution of $4 \times 4$ image patches from the ImageNet32 dataset. For these experiments we learned a Hidden Chow-Liu Tree (HCLT) [Liu and Van den Broeck, 2021] structure over image pixels for the PC, and pixel-level distributions are $1024$-component mixture of fully-factorized distributions on the input channel bits. Surprisingly, in these experiments, test BPD improves when categorical leaves are replaced by Bernoulli distributions. Then, using our initialization technique, the test BPD improves further in the constrained YCoCg-R PC.

## 6 CONCLUSION

In summary, we identify how learning PCs on decorrelated image spaces leaks probability mass to unrealizable values and fails to provide a tractable model on the original space. We use a binary representation of the color channels to encode the transformations, improving modeling performance and providing a tractable model on the original space.

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

# Bitblasting for Constrained Decorrelation in Tractable Image Modeling (Supplementary Material)

**Poorva Garg**[1]     **Benjie Wang**[1]     **Oliver Broadrick**[1]     **Guy Van den Broeck**[1]     **Todd Millstein**[1]

[1]Department of Computer Science, University of California, Los Angeles, California, USA

## A   SMOOTHNESS AND DECOMPOSABILITY

**Definition 2** (Smoothness and Decomposability)**.** *A sum node is* smooth *if all of its children have the same scope. A product node is* decomposable *if its children have disjoint scope.*

*A PC is smooth if all of its sum nodes are smooth.*

*A PC is decomposable if all of its product nodes are decomposable.*

## B   COLOR TRANSFORM DETAILS

We provide additional details of the three color transforms described in the paper below. Once we obtain the transformed channels, they are converted to binary using the standard convention.

**YCoCg-L .** Given integers $R, G, B$ in the range $[0, 255]$, the YCoCg-L transform [Malvar and Sullivan, 2003b] produces values $Y, Co, Cg$ as follows:

$$Y = \frac{R}{4} + \frac{G}{2} + \frac{B}{4}$$
$$Co = \frac{R}{2} - \frac{B}{2}$$
$$Cg = \frac{-R}{4} + \frac{G}{2} + \frac{-B}{4}$$

Note that the above computations results in $Y$ taking values in the interval $[0, 255.75]$ at the granularity of 0.25 thus requiring 10 bits for lossless representation. $Co$ takes values in the interval $[-127.5, 127.5]$ at the granularity of 0.5 requiring 9 bits. And finally, $Cg$ takes values in the interval $[-127.5, 127.5]$ at the granularity of 0.25 requiring 10 bits. Thus, YCoCg-L needs 5 additional bits over RGB.

To avoid $Co$ and $Cg$ taking negative values, they are modeled in a probabilistic circuit only after they are right shifted by adding appropriate values. In particular,

$$\tilde{Co} = Co + 127.5$$
$$\tilde{Cg} = Cg + 127.5$$

**YCoCg-R .** Given integers $R, G, B$ in the range $[0, 255]$, the YCoCg-R transform [Malvar and Sullivan, 2003b] produces

*Accepted for the 8th Workshop on Tractable Probabilistic Modeling at UAI  (TPM 2025).*

values $Y, Co, Cg$, as follows in integer arithmetic:

$$Co = R - B$$
$$tmp = B + \lfloor Co/2 \rfloor$$
$$Cg = G - tmp$$
$$Y = tmp + \lfloor Cg/2 \rfloor$$

where $Y$ takes integer values in $[0, 255]$ and $Co$ and $Cg$ takes integer values in $[-255, 255]$. This transform is exactly invertible (lossless) using only integer arithmetic. Again, note that $Co$ and $Cg$ now require one additional bit in their representation and thus the whole transform need 2 additional bits over RGB.

To avoid $Co$ and $Cg$ taking negative values, they are right shifted by appropriate values as was done in the case of YCoCg-L transform.

$$\tilde{Co} = Co + 255$$
$$\tilde{Cg} = Cg + 255$$

**YCoCg-mod .** Given integers $R, G, B$ in the range $[0, 255]$, the YCoCg-mod transform produces integer values of $Y, Co, Cg$ in the range $[0, 255]$ as described by the following code description.

```
def rgb2yccm(R, G, B):
    def forward_lift(x, y):
        diff = (y - x) % 256
        average = (x + (diff >> 1)) % 256
        return average, diff

    tmp, Co = forward_lift(R, B)
    Y, Cg = forward_lift(G, tmp)

    return Y, Co, Cg
```

Note that YCoCg-mod transform does not require any additional bits over RGB.