# OpenReview forum: "Bitblasting for Tractable Constrained Decorrelation in Image Modeling"
_auai.org/UAI/2025/Workshop/TPM — TPM 2025_

### Official Review · Reviewer_5twn · 2025-06-08
**Bitblasting to demistify bijectivity in image transforms**

**Rating:** 3

**Review:**

The paper tries to demystify the application of RGB image transformations in the context of PC distribution estimation.
Specifically, the paper tackles the problem of “leakage” when applying linear color‐space transforms like YCC.
Authors propose bitblasting, i.e. representing each 8-bit pixel channel as eight Bernoulli bits and encoding the transform as logical constraints via OBDDs such that the learned PC never assigns mass to invalid transformed values.

I find the goal of the paper quite important, however, unfortunately, it reads quite rushed and unclear at times.
Although the idea is promising and well-motivated, the paper would benefit from the inclusion of more concrete details, such as algorithms and toy examples.

some comments/questions/suggestions:
- what's the difference between reversible and bijective? isn't the YCC-R transform a bijection in [0, 255]?
- can you provide python code for all transforms in Table 1?
- OBDD is never formally defined
- provide algorithm to create the OBDDs in Table 2
- during training, can we bijectively transform the data and train without any OBDDs? or are OBDDs needed during training as well?

---

### Official Review · Reviewer_Jyqa · 2025-06-12
**An interesting paper evaluating multiple approaches for channel decorrelation in probabilistic circuits**

**Rating:** 3

**Review:**

The paper proposes using an ordered binary decision diagram (OBDD), i.e., a logical circuit, as a way to leverage decorrelated color spaces (the YCC family) for learning probabilistic circuits (PCs) over image data sets. Crucially, the OBDD is used as a constraint representation to prevent colors in YCC format to be incorrectly associated to some colors in RGB format. In fact, this would otherwise make decorrelation non-bijective and therefore leak probability mass. I think a major issue is that this in turn introduces a bias in experimental results.

I have found the paper easy to follow and with a mostly clear contribution. However, I believe the writing is sometimes imprecise for the following reasons.
- In the paragraph just before Section 4, the authors mention that some colors space transforms leak probability mass _to invalid values_. In my opinion, the values in the transformed color space are still valid in that color space. However, multiple values in the transformed color space correspond to the same RGB value, making the transformation non-bijective, but all color values are still valid. It might be possible I have misunderstood the YCoCg format, but I feel this aspect can be made more clear by making explicit examples with some color values.
- The authors report the bpd results for Categorical input distributions in Tables 3 and 4, with RGB and the YCC color spaces. In my opinion, the way the numbers are made bold is unclear, because the bpds shown e.g. for Categorical-YCCL are not comparable with the bpds for Binary-Constraint-YCCL. The reason is the first bpds leak probability mass, while the second do not. If I understood this aspect correctly, then I suggest the authors to be more explicit that some of these bpds are not comparable for the probability mass leaking issue.

I also found that the YCC-mod colorspace looks already enough to considerably boost the bpds in PCs, even though it has larger pairwise correlations between channels. I guess this weakens the motivation of using YCC-L and YCC-R transforms that instead can suffer from probability mass leaking. However, it might be that for different datesets (e.g., CelebA) this is not the case.

When I was reading the paper, I was thinking about the fact that similar color space transformations can be applied in other generative models as well, e.g., normalizing flows. I think It could be interesting to understand whether these models already implicitly learn these transformed color spaces, while at the same time PCs cannot efficiently learn such transformations from scratch. Therefore, this might be the reason why we observe improvements in distribution estimation with such alternative color spaces.